# Predicting treatment response from longitudinal images using multi-task deep learning

Cheng Jin [1,7], Heng Yu[1,7], Jia Ke[2,3,7], Peirong Ding [4,5,7], Yongju Yi[6], Xiaofeng Jiang[2,3], Xin Duan[2,3], Jinghua Tang[4,5], Daniel T. Chang[1], Xiaojian Wu[2,3✉], Feng Gao[2,3✉] & Ruijiang Li [1✉]

Radiographic imaging is routinely used to evaluate treatment response in solid tumors. Current imaging response metrics do not reliably predict the underlying biological response. Here, we present a multi-task deep learning approach that allows simultaneous tumor segmentation and response prediction. We design two Siamese subnetworks that are joined at multiple layers, which enables integration of multi-scale feature representations and in-depth comparison of pre-treatment and post-treatment images. The network is trained using 2568 magnetic resonance imaging scans of 321 rectal cancer patients for predicting pathologic complete response after neoadjuvant chemoradiotherapy. In multi-institution validation, the imaging-based model achieves AUC of 0.95 (95% confidence interval: 0.91–0.98) and 0.92 (0.87–0.96) in two independent cohorts of 160 and 141 patients, respectively. When combined with blood-based tumor markers, the integrated model further improves prediction accuracy with AUC 0.97 (0.93–0.99). Our approach to capturing dynamic information in longitudinal images may be broadly used for screening, treatment response evaluation, disease monitoring, and surveillance.

[1] Department of Radiation Oncology, Stanford University School of Medicine, Stanford, CA, USA. [2] Department of Colorectal Surgery, The Sixth Affiliated Hospital, Sun Yat-sen University, Guangzhou, China. [3] Guangdong Institute of Gastroenterology, Guangdong Provincial Key Laboratory of Colorectal and Pelvic Floor Diseases, Guangzhou, China. [4] Department of Colorectal Surgery, Sun Yat-sen University Cancer Center, Guangzhou, China. [5] Sun Yat-sen University Cancer Center, State Key Laboratory of Oncology in South China, Collaborative Innovation Center for Cancer Medicine, Guangzhou, China. [6] Center for Network Information, The Sixth Affiliated Hospital, Sun Yat-sen University, Guangzhou, China. [7]These authors contributed equally: Cheng Jin, Heng Yu, Jia Ke, Peirong Ding. ✉email: wuxjian@mail.sysu.edu.cn; gaof57@mail.sysu.edu.cn; rli2@stanford.edu

Accurate prediction of treatment response in individual patients is essential for personalized medicine. Given its noninvasive nature, radiographic imaging is widely used in oncology practice and clinical trials for response evaluation, typically by measuring tumor size change before and after treatment[1]. Because response patterns can be complex and heterogeneous, this simple approach does not always lead to an accurate assessment of the underlying biological response[2]. Despite numerous efforts to improve upon the standard practice, a reliable approach to tumor response prediction remains elusive[3,4].

Deep learning has been extensively used in image analysis for several clinical applications[5]. However, most studies are focused on disease detection and diagnosis[6–12], by analyzing images acquired at one time point during patient care. This approach is inherently limited for response prediction purposes, because it does not take the therapy-induced changes into consideration. Recently, deep learning has been used to analyze longitudinal clinical variables for predicting disease risk or progression[13,14]. Given the special structure of three-dimensional medical image data, however, there remains an unmet need for deep learning methods that effectively extract dynamic information from longitudinal images. Further, it has been challenging to combine tumor segmentation and response prediction, which were traditionally treated as separate problems in medical image analysis. Integration of these interconnected tasks in a unified model may improve the prediction performance.

Globally, more than 700,000 patients are diagnosed with rectal cancer every year[15]. Neoadjuvant chemoradiotherapy (CRT) followed by radical surgery is the standard treatment for locally advanced rectal cancer. Around 15–27% of patients will have a pathologic complete response (pCR), where examination of surgical specimens shows absence of residual cancer cells[16]. Given that surgery is associated with considerable morbidity and poor quality of life, organ-preserving strategies such as "watch-and-wait" are being actively investigated as a non-operative management[17–19]. A prerequisite for clinical implementation of this approach is the accurate prediction of treatment response prior to surgery[20].

Here, we propose a multi-task deep learning approach to predict treatment response and test the model in multi-institution cohorts of rectal cancer patients. The deep neural network performs simultaneously two different but related tasks, i.e., tumor segmentation and response prediction. We show that integration of the two tasks in one network coupled with incorporation of change information in longitudinal images improves accuracy for response prediction.

## Results

**Patients and datasets**. This multi-institution study included patients with locally advanced rectal cancer who were treated with neoadjuvant CRT followed by total mesorectal excision (Fig. 1a). We trained a deep learning model to predict pCR based on pre-treatment and post-treatment MRI and performed independent testing in both internal and external validation cohorts (Fig. 1b). The detailed flowchart for patient enrollment is shown in Supplementary Fig. 1. Specifically, the training cohort consisted of 321 patients who were consecutively treated at a hospital specialized in colorectal disease. After training the model, we prospectively collected data for a cohort of 160 patients treated at the same hospital for internal validation. An independent cohort of 141 patients treated at a second institution was used for external validation.

Patients were classified into two categories according to whether there was a pCR in the resected tumor specimen. For each patient, longitudinal multiparametric magnetic resonance images (MRI) before and after neoadjuvant CRT were collected, including T1-weighted imaging with and without contrast, T2-weighted imaging, and diffusion-weighted imaging (DWI) (Supplementary Table 1). In total, 4976 MRI scans from 622 patients were analyzed.

The patient characteristics of the three cohorts are summarized in Table 1. We compared the distribution of different clinical variables between the two response groups (pCR vs. non-pCR). Neither demographic (age, gender) nor pre-treatment disease characteristics (tumor location, T stage, N stage) was consistently associated with pCR across different cohorts. However, there was a statistically significant association between pCR and post-treatment T stage ($p < 0.001$) in all three cohorts. This suggests, not surprisingly, that radiologic evaluation of the tumor on post-treatment imaging may be a better indicator of treatment response than on baseline imaging. In terms of prediction for pCR, post-treatment T stage (T0 vs. T1–4) had a good accuracy of 88%, 81%, and 86% but a moderate positive predictive value of 61%, 62%, and 69% in the training, internal, and external validation cohorts, respectively.

**Proposed network model**. In order to effectively capture the dynamic information contained in longitudinal images, we proposed a multi-task learning framework with a deep neural network architecture (3D RP-Net). The network consists of two main components: (1) a convolutional encoding/decoding subnetwork for feature extraction and tumor segmentation, and (2) a multi-stream Siamese subnetwork for response prediction (Fig. 2a). The feature extraction and segmentation subnetwork consists of two identical 3D U-net with shared parameters. The response prediction subnetwork combines the extracted image features from three different network layers via depth-wise convolution (Fig. 2b). This allows for integration of multi-scale feature representations and comprehensive pair-wise comparison between images at the two time points. More details about the network design are presented in "Methods" and Supplementary Fig. 2.

**Model performance**. After training the network, we prospectively collected data for additional patients from two institutions and independently tested this model. The tumor segmentation from the proposed network was in good agreement with expert delineation, and the results were very similar to specialized deep neural networks trained with a single task, i.e., tumor segmentation (Supplementary Fig. 3). For response prediction, the proposed 3D RP-Net achieved consistently high accuracy across the training and two validation cohorts (Fig. 3a). The AUC was 0.95 (95% CI: 0.91–0.98) and 0.92 (95% CI: 0.87–0.96) in the internal and external validation cohorts, respectively. At the optimal cutoff point, the 3D RP-Net showed sensitivity at 93% and 91%, specificity at 94% and 92% for predicting pCR for the two validation cohorts (Fig. 3e).

**Improvement over models with pre- or post-treatment image or T2-weighted image alone**. To demonstrate the importance of including dynamic information for response evaluation, we compared 3D RP-Net with deep neural networks trained with either pre-treatment or post-treatment multiparametric MRI alone. In both validation sets, the proposed network that incorporates changes before and after treatment significantly improved the AUC by 14–19% in absolute terms (all $p < 0.01$) compared with ResNet-18 models trained using image at one time point only (Fig. 3b, c, and Supplementary Tables 2 and 3).

**a**

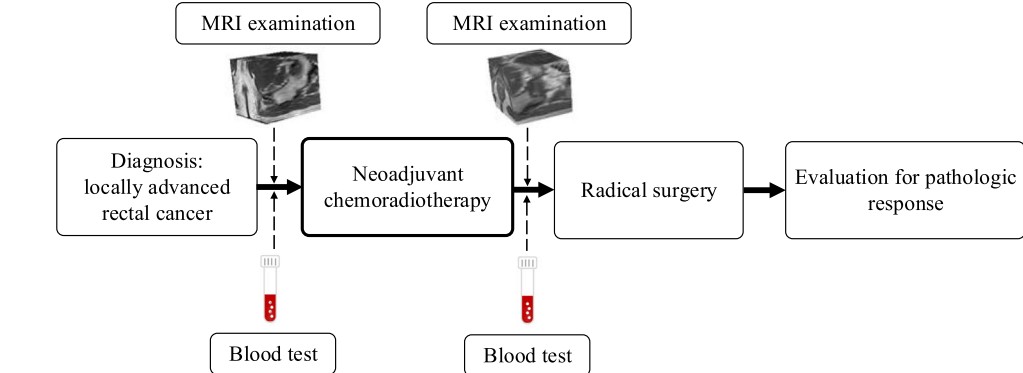

**b**

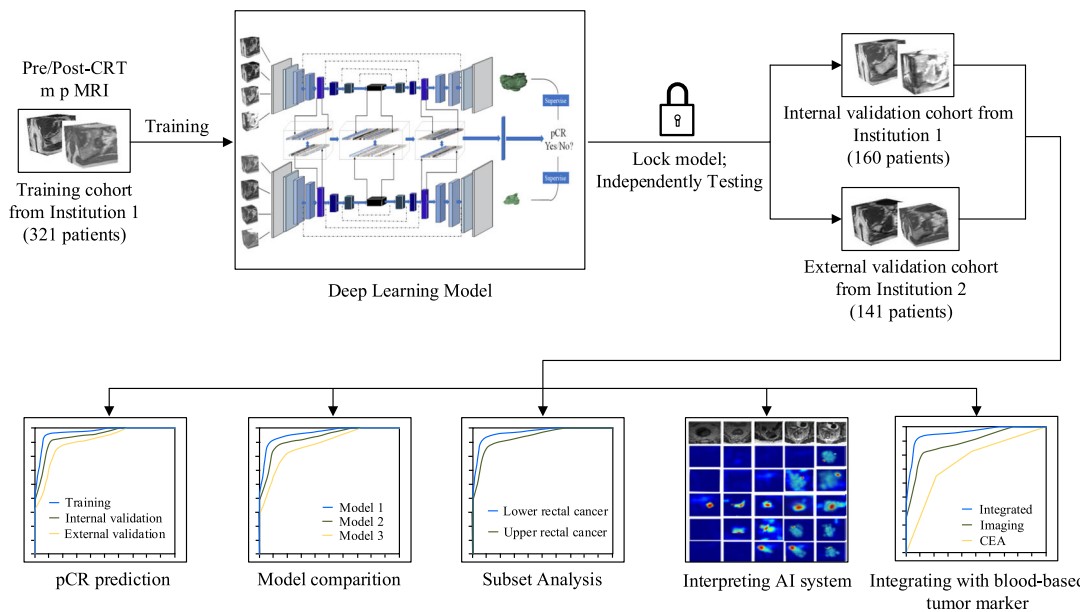

**Fig. 1 Clinical workflow and study design. a** Diagnosis, treatment, and response evaluation for patients with rectal cancer. MRI examinations and blood tests were performed for each patient before and after neoadjuvant chemoradiotherapy. **b** Development and validation of a deep learning system to predict pathologic complete response from longitudinal imaging.

We then assessed the response prediction performance by training a network model using T2-weighted imaging only, which is the most commonly used imaging technique in rectal cancer. Although the prediction using pre-treatment and post-treatment T2-weighted imaging was quite accurate with AUC 0.86–0.88 in the validation cohorts (Supplementary Table 4), it did not outperform the model trained using multiparametric imaging, which indicates the benefit of incorporating complementary anatomic and functional imaging information.

**Improvement over alternative deep learning models**. We then assessed how the proposed multi-task approach compared with alternative deep learning using the longitudinal multiparametric MRI for response prediction. For comparison, Siamese networks with a ResNet were trained for a single task of predicting response. Two types of network schemes were adopted: (1) a traditional Siamese network where two subnetworks are joined at the fully connected layer with simple concatenation; (2) an improved Siamese network where two subnetworks are joined at

three different layers, similarly as our network. Both comparison networks are single-task learning, with or without multi-scale feature integration (Supplementary Fig. 4). We also compared with traditional radiomics models (Supplementary Methods).

In both internal and external validation sets, the proposed network again achieved better performance for response prediction compared with single-task learning (Fig. 3b, c, and Supplementary Tables 2 and 3). The AUC was improved in absolute terms by 7% and 8% ($p < 0.01$, adjusted for multiple comparison) in the internal and external validation cohorts, respectively. The proposed deep learning model also outperformed the radiomics model for pCR prediction (Supplementary Table 5). These results confirm that the integration of tumor segmentation in the multi-task learning framework is crucial for its superior response evaluation. Interestingly, we note that for single-task learning, multi-scale feature integration also improved performance over simple concatenation used in classical Siamese networks (Supplementary Fig. 5), indicating its beneficial role in effectively mining change information in longitudinal images.

**Table 1 Patient characteristics in the training, internal validation, and external validation cohorts.**

| Characteristics | Training cohort | | | Internal validation cohort | | | External validation cohort | | |
|---|---|---|---|---|---|---|---|---|---|
| | Non-pCR (n = 264) | pCR (n = 57) | p | Non-pCR (n = 116) | pCR (n = 44) | p | Non-pCR (n = 98) | pCR (n = 43) | p |
| Gender | | | <0.001 | | | 0.011 | | | 0.157 |
| Female | 37 (14.0%) | 17 (29.8%) | | 36 (31.0%) | 5 (11.4%) | | 27 (27.6%) | 17 (39.5%) | |
| Male | 227 (86.0%) | 40 (70.2%) | | 80 (69.0%) | 39 (88.6%) | | 71 (72.4%) | 26 (60.5%) | |
| Age, mean ± SD, years | 53.6 ± 12.1 | 51.9 ± 12.8 | 0.202 | 54.6 ± 11.1 | 52.1 ± 13.0 | 0.124 | 56.3 ± 11.1 | 50.2 ± 9.2 | <0.001 |
| Tumor location | | | 0.011 | | | 0.638 | | | 0.844 |
| Upper | 33 (12.5%) | 5 (8.8%) | | 9 (7.8%) | 3 (6.8%) | | 11 (11.2%) | 4 (9.3%) | |
| Upper middle | 50 (18.9%) | 7 (12.3%) | | 6 (5.2%) | 0 (0%) | | 5 (5.1%) | 1 (2.3%) | |
| Middle | 41 (15.5%) | 4 (7.0%) | | 29 (25.0%) | 14 (31.8%) | | 42 (42.8%) | 16 (37.2%) | |
| Middle lower | 52 (19.7%) | 7 (12.3%) | | 5 (4.3%) | 2 (4.5%) | | 4 (4.1%) | 2 (4.7%) | |
| Lower | 88 (33.3%) | 34 (59.6%) | | 67 (57.8%) | 25 (56.8%) | | 29 (29.6%) | 10 (23.3%) | |
| Pre-CRT T stage | | | 0.826 | | | 0.374 | | | 0.179 |
| T1 | 9 (3.4%) | 2 (3.5%) | | 1 (0.9%) | 0 (0%) | | 1 (0.1%) | 0 (0%) | |
| T2 | 12 (4.5%) | 4 (7.0%) | | 3 (2.6%) | 2 (4.5%) | | 4 (4.1%) | 3 (7.0%) | |
| T3 | 185 (70.1%) | 41 (71.9%) | | 91 (78.4%) | 31 (70.5%) | | 81 (82.7%) | 30 (69.8%) | |
| T4a | 40 (15.2%) | 6 (10.5%) | | 9 (7.8%) | 2 (4.5%) | | 4 (4.1%) | 1 (2.3%) | |
| T4b | 18 (6.8%) | 4 (7.0%) | | 12 (10.3%) | 9 (20.5%) | | 8 (8.2%) | 9 (20.9%) | |
| Pre-CRT N stage | | | 0.419 | | | 0.061 | | | 0.02 |
| N0 | 56 (21.2%) | 13 (22.8%) | | 27 (23.3%) | 10 (22.7%) | | 23 (23.5%) | 9 (20.9%) | |
| N1a | 69 (26.1%) | 8 (14.0%) | | 16 (13.8%) | 13 (29.5%) | | 12 (12.2%) | 12 (27.9%) | |
| N1b | 36 (13.6%) | 7 (12.3%) | | 23 (19.8%) | 2 (4.5%) | | 19 (19.4%) | 2 (4.7%) | |
| N1c | 8 (3.0%) | 4 (7.0%) | | 7 (6.0%) | 1 (2.3%) | | 8 (8.2%) | 2 (4.7%) | |
| N2a | 66 (25.0%) | 17 (29.8%) | | 32 (27.6%) | 14 (31.8%) | | 33 (33.7%) | 13 (30.2%) | |
| N2b | 29 (11.0%) | 8 (14.0%) | | 11 (9.5%) | 4 (9.1%) | | 3 (3.1%) | 5 (11.6%) | |
| Pre-CRT CRM | | | 0.257 | | | <0.001 | | | 0.521 |
| Negative | 215 (81.4%) | 50 (87.7%) | | 95 (81.9%) | 37 (84.1%) | | 75 (76.5%) | 35 (81.4%) | |
| Positive | 49 (18.6%) | 7 (12.3%) | | 21 (18.1%) | 7 (15.9%) | | 23 (23.5%) | 8 (18.6%) | |
| Post-CRT T stage | | | <0.001 | | | <0.001 | | | <0.001 |
| T0 | 30 (11.4%) | 47 (82.5%) | | 22 (19.0%) | 36 (81.8%) | | 19 (19.4%) | 42 (97.7%) | |
| T1 | 11 (4.2%) | 6 (10.5%) | | 21 (18.1%) | 4 (9.1%) | | 18 (18.4%) | 1 (2.3%) | |
| T2 | 107 (40.5%) | 3 (5.2%) | | 48 (41.3%) | 3 (6.8%) | | 38 (38.8%) | 0 (0%) | |
| T3 | 51 (19.3%) | 1 (1.8%) | | 15 (12.9%) | 1 (2.3%) | | 12 (12.2%) | 0 (0%) | |
| T4a | 45 (17.0%) | 0 (0%) | | 4 (3.4%) | 0 (0%) | | 5 (5.1%) | 0 (0%) | |
| T4b | 20 (7.6%) | 0 (0%) | | 6 (5.2%) | 0 (0%) | | 6 (6.1%) | 0 (0%) | |
| Post-CRT N stage | | | <0.001 | | | 0.104 | | | 0.068 |
| N0 | 174 (65.9%) | 52 (91.2%) | | 86 (74.1%) | 42 (95.5%) | | 80 (81.7%) | 43 (100%) | |
| N1a | 52 (19.7%) | 5 (8.8%) | | 16 (13.8%) | 2 (4.5%) | | 9 (9.2%) | 0 (0%) | |
| N1b | 10 (3.8%) | 0 (0%) | | 6 (5.2%) | 0 (0%) | | 5 (5.1%) | 0 (0%) | |
| N1c | 8 (3.0%) | 0 (0%) | | 4 (3.4%) | 0 (0%) | | 2 (2.0%) | 0 (0%) | |
| N2a | 15 (5.7%) | 0 (0%) | | 3 (2.6%) | 0 (0%) | | 1 (1.0%) | 0 (0%) | |
| N2b | 5 (1.9%) | 0 (0%) | | 1 (0.9%) | 0 (0%) | | 1 (1.0%) | 0 (0%) | |
| Post-CRT CRM | | | 0.004 | | | 0.115 | | | 0.006 |
| Negative | 224 (84.8%) | 56 (98.2%) | | 103 (88.8%) | 43 (97.7%) | | 83 (84.7%) | 43 (100%) | |
| Positive | 40 (15.2%) | 1 (1.8%) | | 13 (11.2%) | 1 (2.3%) | | 15 (15.3%) | 0 (0%) | |

Data shown are the number and percentage of patients, with the exception of age (mean and SD). Statistical comparisons were performed for each clinical variable between the two response groups (pCR vs. non-pCR). p values were computed using the two-sided t test for age as a continuous variable and the Chi-square test or Fisher's exact test for categorical variables, as appropriate. Stage and CRM status was assessed by magnetic resonance imaging. pCR pathologic complete response, SD standard deviation, CRT chemoradiotherapy, CRM circumferential resection margin.

**Model performance in patient subgroups by tumor location, gender, magnetic field strength.** Depending on the tumor location along the rectum, which can extend more than 10 cm, tumor response to systemic therapy and surgical management may be different. We thus divided the patients into two subsets of upper/middle and lower rectal cancer and evaluated the network performance separately. The 3D RP-Net obtained satisfactory prediction in both patient subgroups, with slightly better performance in lower rectal cancer with AUC of 0.95 (Fig. 3d and Supplementary Table 6). Results were similar between internal and external validation cohorts (Supplementary Fig. 6 and Supplementary Table 7). In addition, we evaluated the model performance in patients within each gender, and the results were slightly better in the male gender (Supplementary Tables 8 and 9).

We also performed subgroup analysis based on the magnetic field strength (Supplementary Tables 10 and 11). There was a slight improvement in prediction performance (around 2–4% increase in AUC) in patients scanned under a magnetic field strength of 3 T compared with 1.5 T. This is possibly because of the improved image quality in 3 T MRI.

**Model calibration and association with tumor regression grade.** In addition to discrimination, we also assessed the model calibration. The proposed 3D RP-Net showed good calibration with close agreement with the observed probabilities of pCR at both ends of the calibration curve (Supplementary Fig. 7). Beyond the binary classification for pCR, we also assessed the relation

**a**

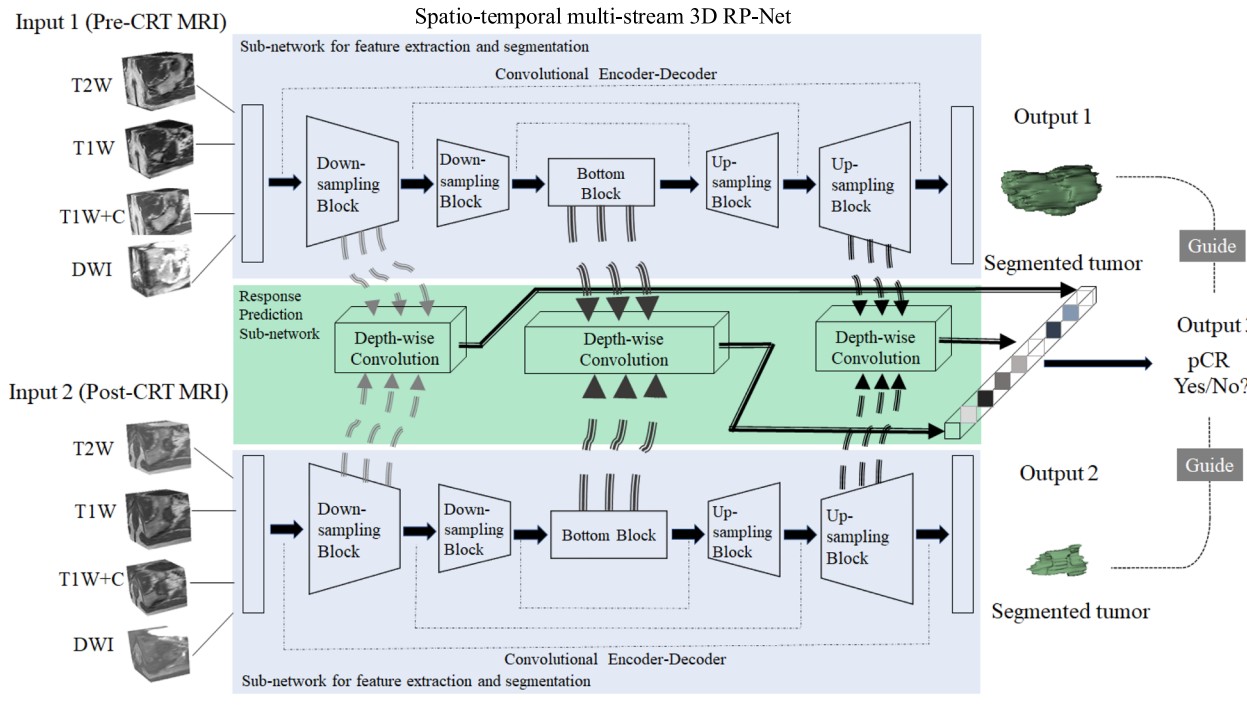

**b**

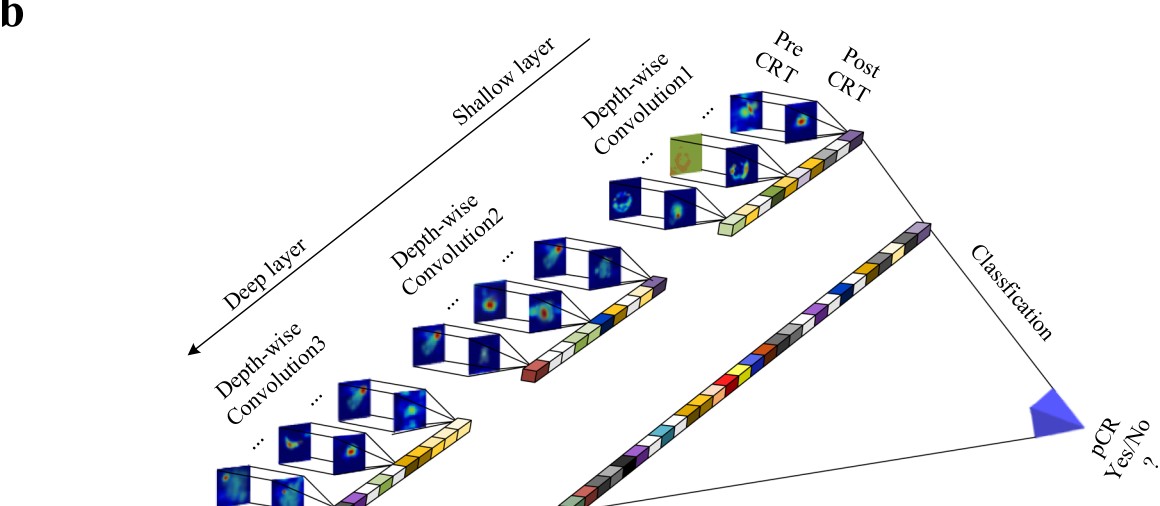

**Fig. 2 Proposed network model for response prediction (3D RP-Net). a** The multi-task deep learning network consists of two subnetworks: one for feature extraction and tumor segmentation, and one for response prediction. The network takes pre- and post-therapy images as inputs and performs two tasks simultaneously: tumor segmentation and response prediction. **b** Depth-wise convolution of pre- and post-therapy images at multiple network layers for multi-scale feature integration and response prediction.

between four categories of tumor regression grade (TRG) and established imaging parameters as well as the deep learning model. While there was a general trend for MRI-defined tumor shrinkage across TRG, the change in tumor volume exhibited substantial variability and overlap between TRG0 (pCR) and TRG1 (non-pCR) and the difference was not statistically significant in the internal validation cohort (Supplementary Fig. 8). On the other hand, each of the four TRG was significantly associated with the deep learning score ($p < 0.01$, adjusted for multiple comparison), with a consistent pattern across different cohorts (Supplementary Fig. 9). In particular, the TRG0 and TRG1 groups were well separated based on the deep learning score (Supplementary Fig. 9).

**Visualization and model interpretation.** Next we sought to understand which areas of the image and what kind of features contributed to the network's output. First, we note that both morphological and physiological information contained in multiparametric MRI is useful for response prediction, and the two types of information tend to be captured in different layers of the network. The image features at shallow layers mainly reflect the structural information, such as tumor boundary, shape, and texture, based on T1w and T2w MRI. On the other hand, features at deep layers mainly represent high-level semantic tumor characteristics from anatomical images as well as functional information contained in DWI (Supplementary Fig. 10).

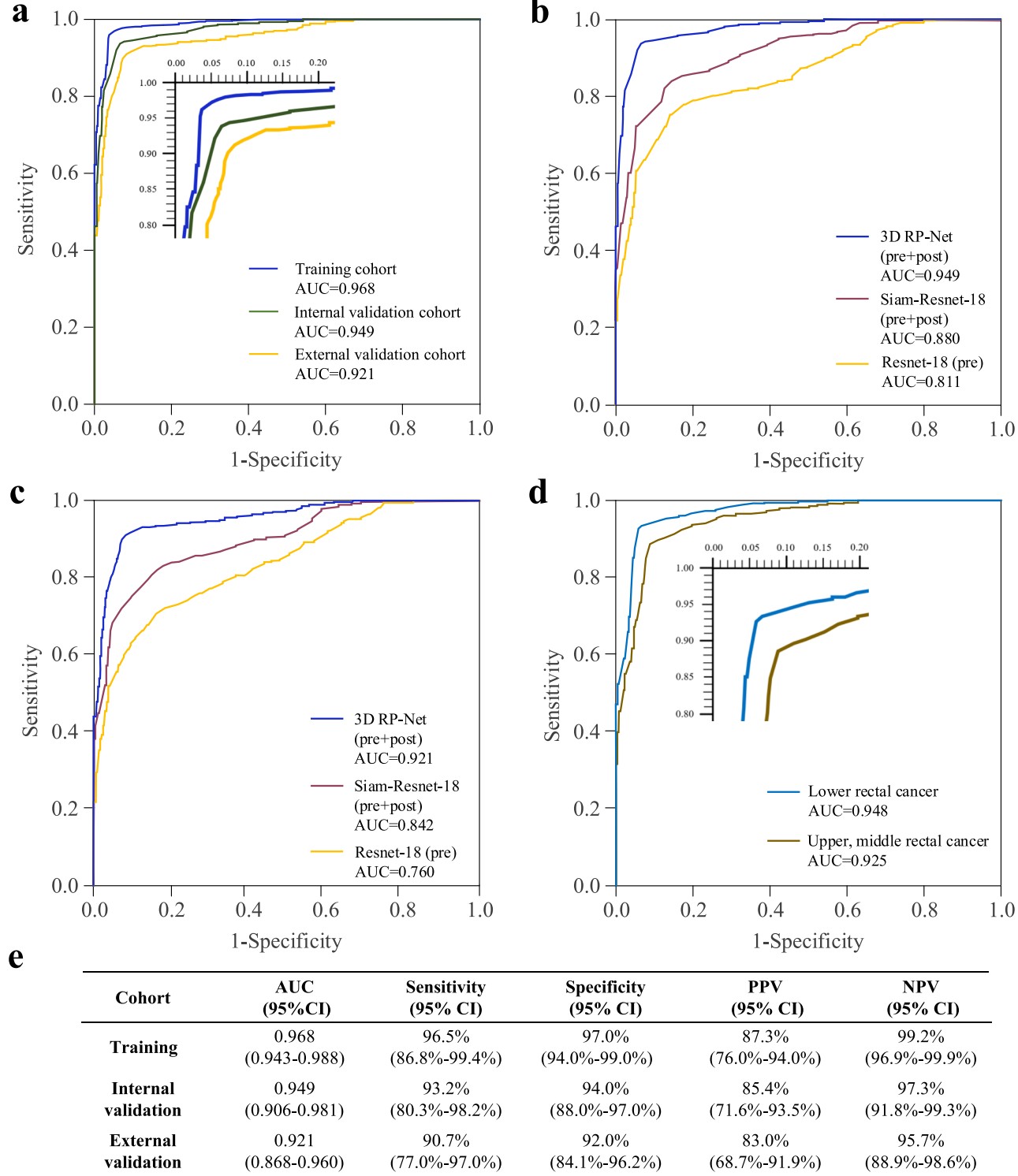

**Fig. 3 Performance for predicting pathologic complete response. a** Receiver operating characteristic (ROC) curves of the proposed 3D RP-Net in the training and two validation cohorts. **b** ROC curves of three different network models in the internal validation cohort. **c** same as (**b**), except for external validation cohort. **d** ROC curves in the subgroup of patients with upper, middle, and lower rectal cancer in the internal validation cohort. **e** Detailed information for prediction performance of the proposed model in the study cohorts. AUC, area under the ROC curve; PPV, positive predictive value; NPV, negative predictive value.

Another notable property is that because of depth-wise convolution in the response prediction subnetwork, the feature maps for the channels in each layer are nearly orthogonal to each other. This allows for the activation of only a few channels in each layer leading to sparse feature maps. Similar features tend to be activated in the same channel, and this pattern was consistently observed across patients. For example, among all 256 channels in the intermediate layer of the response prediction subnetwork, less than 10 channels showed high activation. These feature maps were evaluated by the radiologist in conjunction with the original multiparametric MRI,

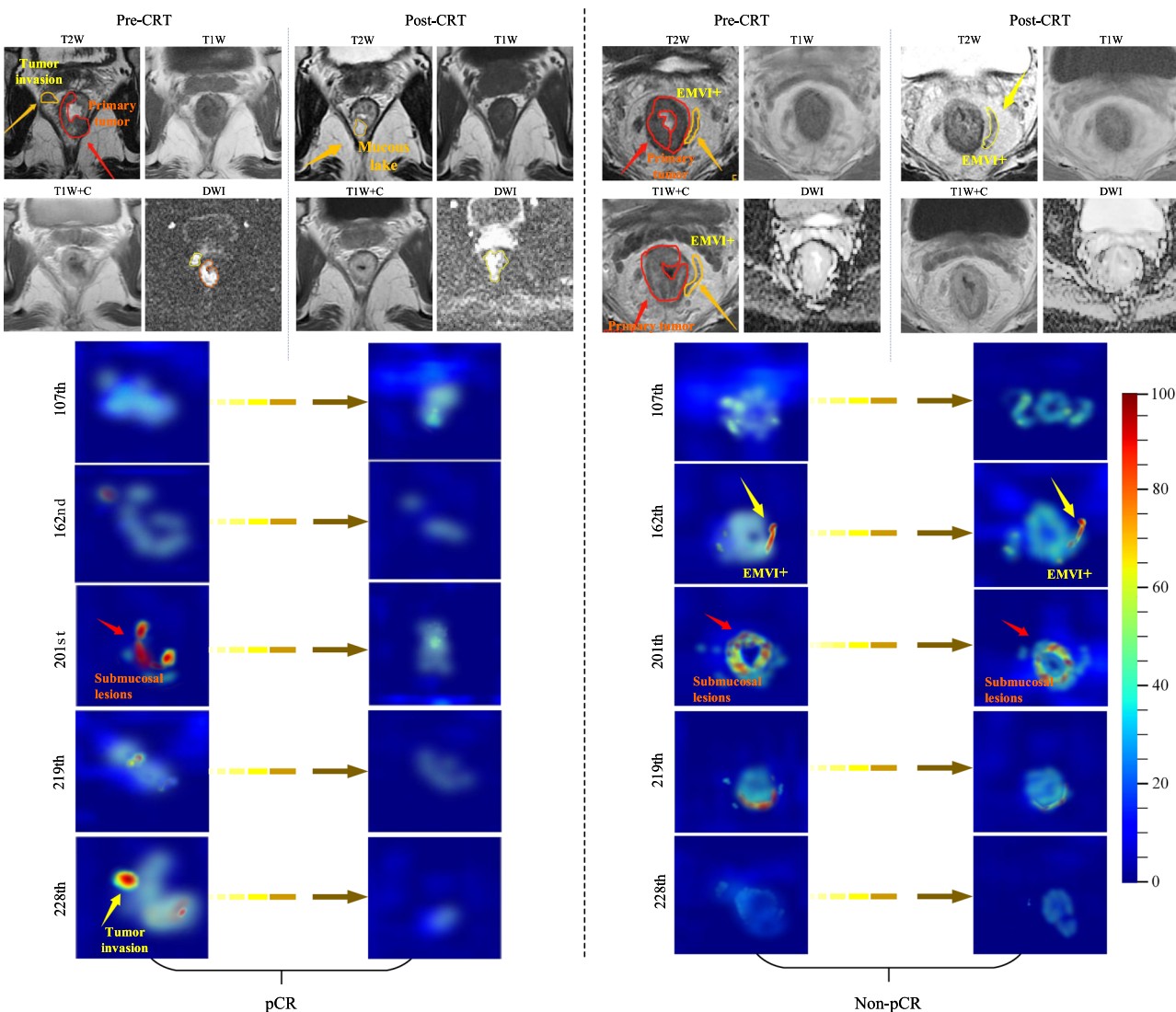

**Fig. 4 Network visualization and interpretation.** Pre- and post-therapy MRI and the corresponding feature map visualization in key channels at the bottom of 3D RP-Net for two representative patients who were correctly predicted to be pCR and non-pCR, respectively. In the 107th channel corresponding to positive lymph nodes, there was no activation (maximum magnitude <50%) in either case. In the 162nd channel corresponding to EMVI, feature map was not activated in the case with pCR, while the overall magnitude decreased by only 9.2% ($p = 0.89$) with non-pCR. In the 201st channel corresponding to submucosal lesions, the overall magnitude of the feature map decreased by 88.9% ($p < 0.001$) with pCR, but decreased by only 11.3% ($p = 0.67$) with non-pCR. In the 219th channel corresponding to mesorectum invasion, the overall magnitude of the feature map decreased by 85.7% ($p < 0.001$) with pCR, but decreased by only 16.7% ($p = 0.41$) with non-pCR. In the 228th channel corresponding to tumor invasion, the overall magnitude of the feature map decreased by 90.5% ($p < 0.001$) with pCR, while tumor invasion was not activated with non-pCR. $p$ values were computed based on the two-sided paired $t$ test between the corresponding feature maps within each channel ($n = 256$ feature values) and adjusted for multiple comparisons. CRT, chemoradiotherapy; pCR, pathologic complete response; EMVI, extramural vascular invasion.

and were found to be related to pathophysiologic characteristics such as mesorectum invasion, extramural vascular invasion, and lymph node involvement (Supplementary Fig. 11).

We selected two representative patients in the external validation set, one with pCR and one with non-pCR. The feature maps in key channels of the response prediction subnetwork were quite similar in the baseline pre-treatment MRI, as shown in Fig. 4. On the other hand, there were substantial decreases of these salient features from pre- to post-therapy images for the patient with pCR. For instance, the overall magnitude of the feature map decreased by 88.9% between the two images (paired $t$ test $p < 0.001$, adjusted for multiple comparison) in the 201st channel related to submucosal lesions. By contrast, for the patient with non-pCR, these changes were mostly minor or modest, with only 11.3% decrease ($p = 0.64$) in the same channel.

**Integration with blood-based biomarkers**. Finally, considering that imaging mainly captures local tumor response while blood-based markers better reflect systemic disease, we combined the imaging model with dynamic changes of blood CEA levels in an integrated model. Based on the pre- and post-CRT CEA levels and change information, we defined five discrete categories of patients with varying degrees of response (Fig. 5a). The CEA model alone had a moderate accuracy for predicting pCR. When combined with imaging, the integrated model achieved the highest AUC 0.97 (95% CI: 0.93–0.99) among three models in the validation cohort (Fig. 5b). The deep learning-based imaging score played a dominant role in the integrated model (Supplementary Fig. 12). Incorporation of post-therapy CEA level also improved imaging model performance but did not outperform the dynamic model (Supplementary Fig. 13 and Supplementary

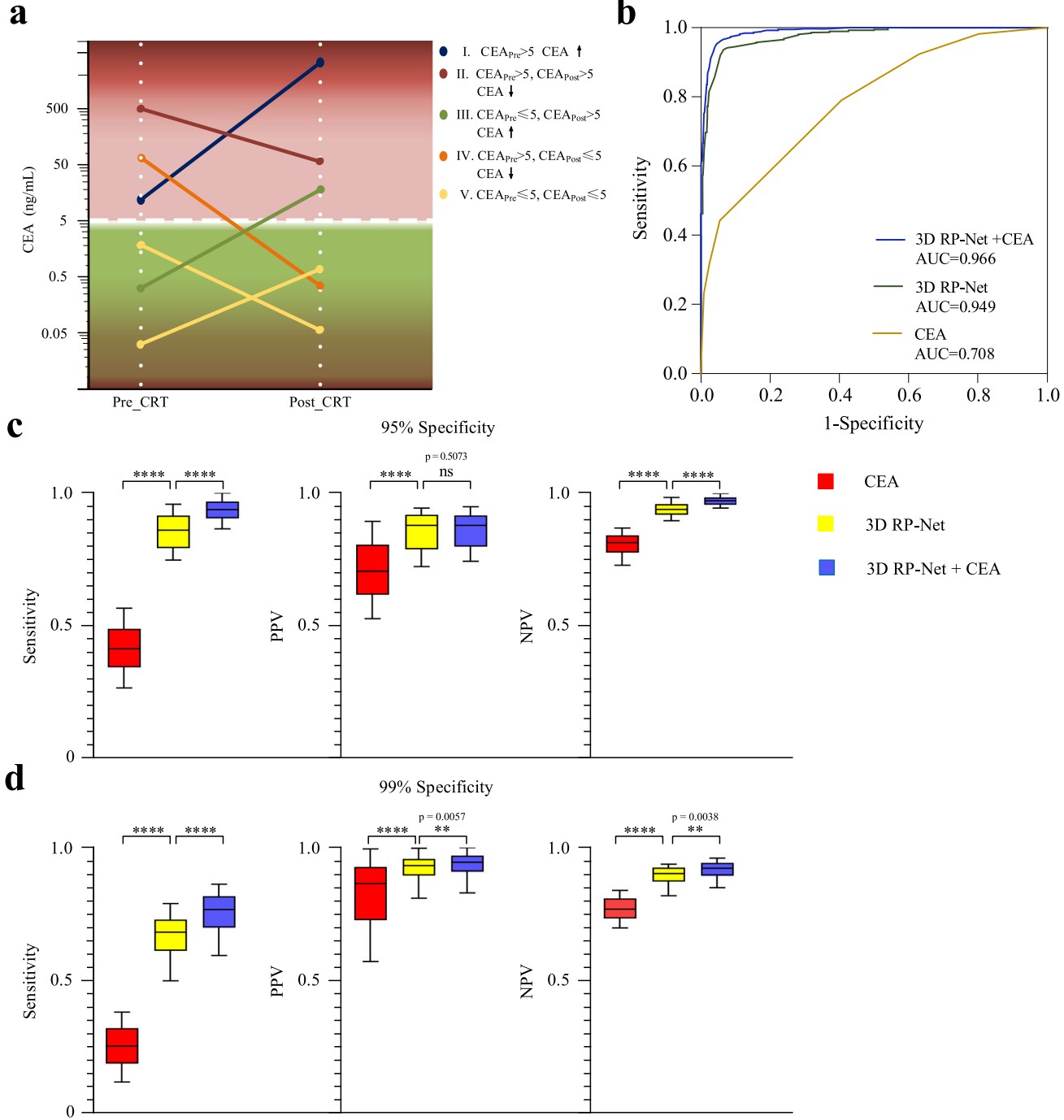

**Fig. 5 Integration of imaging and blood-based biomarkers. a** Definition of five discrete categories of blood marker response based on the clearance patterns of CEA level before and after CRT. **b** ROC curves for three different models: combined imaging and CEA model, imaging alone, and CEA alone in the internal validation cohort. **c** Comparison of response prediction performance of different models at 95% specificity. **d** same as (**c**) except for 99% specificity. In the box plots, the central line represents the median, the bounds of box correspond to the first and third quartiles, and the whiskers are the minimum and maximum of the data. $p$ values were computed based on the two-sided $t$ test ($n = 160$ patients) between the prediction models as indicated in (**c**, **d**) and adjusted for multiple comparisons. CEA, carcinoembryonic antigen; PPV, positive predictive value; NPV, negative predictive value. ns, not significant, $p \geq 0.05$; [*]$0.01 \leq p < 0.05$; [**]$0.001 \leq p < 0.01$; [***]$0.0001 \leq p < 0.001$; [****]$p < 0.0001$.

Table 12). At different specificity thresholds, the integrated model significantly improved sensitivity and negative predictive value ($p < 0.01$, adjusted for multiple comparison) over the imaging model (Fig. 5c, d). Of note, the integrated model maintained relatively high sensitivity at 99% specificity, and the positive predictive value surpassed 97%, meaning that only 3% patients

eligible for watchful waiting under this model would have residual disease after neoadjuvant therapy.

## Discussion
In this work, we present a multi-task deep learning approach to predict tumor response by leveraging dynamic information

contained in longitudinal images. The proposed deep neural network achieved accurate prediction of pCR to neoadjuvant CRT in rectal cancer. This may help identify which patients will have no residual cancer after neoadjuvant CRT and can safely undergo watchful waiting, avoiding potentially serious complications from radical surgery.

The multi-task learning approach allows the network to perform tumor segmentation and response prediction simultaneously. The performance of the proposed network for tumor segmentation was on par with specialized deep neural networks trained with a single task. Clinically, tumor delineation plays an important role in surgical planning and radiation treatment planning. However, reliable tumor segmentation is challenging, limited by intra/inter-rater variations even among expert physicians. The multi-task learning approach may be used to generate more consistent tumor contours and can benefit several clinical applications.

Because our primary goal is to predict response, one key question is whether tumor segmentation is truly necessary or simply a by-product of the network. By comparing with networks that do not perform explicit segmentation, we showed that the segmentation subnetwork was actually a critical component of the model and integration of the two subnetworks led to superior accuracy in response prediction. One reason could be that tumor boundary information provided by segmentation enables the network to focus on the most relevant regions for response prediction. Indeed, image features of the peritumoral region have been associated with key aspects of tumor biology and clinical outcome[21–24].

An important distinction of our study from previous works is the incorporation of longitudinal imaging for response prediction. There has been intensive investigation on the use of baseline pre-treatment images for predicting (mainly survival) outcomes of cancer patients[4]. This approach is fundamentally limited by the inability to incorporate information about tumor changes caused by treatment. Although tumor phenotypes can have substantial variations, there may be shared imaging characteristics in the way it responds to treatment, such as change in tumor size, which is the basis for response evaluation in clinical practice. In addition, changes in tumor cellular density and blood perfusion are common after CRT. These changes can be reflected on multi-parametric MRI that provides complementary anatomic and functional information.

In order to effectively mine these treatment-induced changes, we designed two Siamese subnetworks with pre/post-therapy images as input. We emphasize that our network is designed not only to learn features of the tumor itself, but also the dynamic changes in response to therapy. Network visualization revealed several high-risk features including depth of tumor invasion and extramural vascular invasion, which were associated with poor response, consistent with previous findings[25,26]. Importantly, we observed substantial decreases in these features from pre- to post-therapy images in responders compared with non-responders, while these feature maps were quite similar in the pre-treatment images. This reinforces the notion that information about tumor response is mainly contained in the change of imaging phenotypes before and after treatment, confirming the effectiveness of our approach.

Radiomics has been used to predict pathologic response after neoadjuvant CRT in rectal cancer. These studies included small, single-institution cohorts, mostly using pre-treatment images[27–30]. In a recent multi-institution study, Liu et al. used radiomic analysis of pre-treatment T2-weighted MRI and DWI to predict distant metastasis after surgery in rectal cancer[31]. The radiomics approach relies on domain expertise to manually define hand-crafted features. Radiomics also requires

accurate tumor segmentation, which can be challenging in practice. By contrast, our multi-task deep learning approach allows both precise tumor segmentation and accurate response prediction.

Although deep learning has been widely used for disease detection and diagnosis, there is a paucity of methods that are designed to track disease progression in longitudinal data[13,14]. Recently, this has been explored for monitoring the natural history of disease[10,32] or assessing response to treatment with longitudinal imaging[33]. However, from a technical perspective, there are some key limitations with previous approaches. First, these networks were designed for the single task of risk prediction without explicit segmentation of the disease. Second, previous studies adopted the classical Siamese structure, where two sub-networks were joined at the final output layer of the network. Consequently, comparison between two images was only possible at the highest abstraction level, which significantly limits the amount of information that can be extracted. In this work, we designed the network by combining feature representations from multiple (shallow, intermediate, deep) layers, which allowed for multi-scale integration and in-depth comparison of paired images. Indeed, our results from ablation studies confirm that multi-task learning coupled with multi-scale integration achieved superior prediction performance compared with single-task learning with or without multi-scale integration. Finally, different from previous single-institution studies using deep learning[33], we conducted rigorous internal and external validation of our model in multi-institution cohorts.

By integrating imaging with complementary blood-based markers, we show that the model performance could be further improved. Here, we used blood CEA level which is an established marker of response in rectal cancer[34,35]. Beyond traditional protein markers, other analytes such as circulating tumor DNA are being investigated for liquid biopsy[36] and may also be used in combination with imaging to further improve response prediction.

This study has some limitations. First, it is a retrospective study and subject to potential selection bias. The generalizability and clinical utility of the proposed model should be rigorously tested in future prospective studies. Second, our deep learning model was trained using data from Asian patients, and its reproducibility across different ethnic groups such as patients from Western populations remains to be evaluated. Finally, in order to select patients who can safely forego surgery, the positive predictive value for complete response must be very high. This deep learning model is not yet ready for clinical use, given the need for prospective validation and demonstration of a sufficiently high positive predictive value. In future, it will be important to integrate information from other investigations such as clinical examination, endoscopic assessment, or molecular approaches to further improve the prediction accuracy.

While the current study is focused on analysis of multi-parametric MRI, the network can also take input from multi-modality imaging such as PET/MRI and PET/CT, which reveal additional aspects of tumor biology such as metabolism. With minor modifications, the network can be adapted to incorporate imaging data acquired at multiple time points, which will open the door to many other clinical applications including cancer screening, monitoring of treatment response and resistance, and surveillance for disease relapse.

In conclusion, we present a multi-task deep learning approach to predict tumor response by extracting treatment-induced change information from longitudinal images. Our approach can be used to improve treatment response evaluation with the potential to inform personalized treatment.

## Methods

**Study design and patients**. We collected clinical, pathologic, and imaging data for rectal cancer patients enrolled from two institutions. The inclusion criteria were: (1) pathologically confirmed diagnosis of locally advanced rectal adenocarcinoma; (2) treatment with neoadjuvant CRT followed by total mesorectal excision; (3) both pre-CRT and post-CRT MRI scans available within one week prior to the initiation of CRT and surgery, respectively. The exclusion criteria were: (1) patients had other concurrent malignancies or had previously received anticancer treatment; (2) patients did not complete the entire course of CRT or did not undergo radical surgery. Patients were also excluded if the MRI quality was insufficient or relevant clinical and pathologic information was missing or incomplete.

In total, 622 patients were included in this retrospective study. The training cohort consisted of 321 patients who were treated from 2013 to 2016 at the Sixth Affiliated Hospital (SAH), Sun Yat-sen University, Guangzhou, China. An independent cohort of 160 patients treated from 2017 to 2018 at the same hospital was used for validation purposes. In addition, an external validation cohort of 141 patients was enrolled at the Sun Yat-sen University Cancer Center (SYSUCC), Guangzhou, China. The detailed flowchart for patient enrollment is shown in Supplementary Fig. 1.

This study was approved by the institutional review boards at the respective institutions (SAH and SYSUCC, Guangzhou) and was conducted in accordance with ethical standards of the Helsinki Declaration. Informed consent was waived for this retrospective study, as no protected health information was used.

**Definition of pathologic response**. Information about pathologic response to neoadjuvant CRT was obtained through detailed histopathological analysis of the resected tumor specimen. The TRG was defined according the AJCC system[37]. TRG 0 indicates complete regression, with no viable tumor cells remaining in the specimens; TRG 1 indicates near-complete regression with single or small number of tumor cells; TRG 2 indicates moderate regression with residual cancer outgrown by fibrosis; TRG 3 indicates minimal or no regression. Only patients with a pCR would be candidates eligible for watchful waiting (without the need for radical surgery). Therefore, patients were divided into two main groups: pCR (TRG 0) and non-pCR (TRG 1–3).

**Image acquisition and processing**. All patients underwent multiparametric MRI scans before and after neoadjuvant CRT. MRI sequences included T1-weighted imaging with and without Gadolinium contrast, T2-weighted imaging, and DWI. For each patient, the pre- and post-CRT images were spatially aligned in 3D using rigid registration given the tumor location in pelvis. Registration was visually checked by anatomical landmarks such as bony structures. Rather than simply stacking the multiparametric images together, we combined them in a four-dimensional tensor image with the last dimension being each of the four MRI sequences and fed this as input to the network model. Details regarding the MRI acquisition protocol and image processing can be found in Supplementary Methods and Supplementary Table 1. Some differences in imaging protocol across institutions and residual registration errors exist as expected. These issues were addressed by data normalization and harmonization, as well as data augmentation techniques during network training.

**Development of the deep learning model**. The proposed multi-task learning network takes pre- and post-therapy multiparametric MRI as input, and outputs are both tumor segmentation and response prediction. The network architecture consists of two subnetworks: one for feature extraction and tumor segmentation, and one for response prediction (Fig. 2a). The segmentation subnetwork consists of two identical 3D U-net, which contains a contracting path, an expansive path, and skip connections between the corresponding layers. The Siamese subnetwork for response prediction combines the extracted image features via depth-wise convolution from three distinct network layers: (1) intermediate layer in the contracting path, (2) bottom layer of the U-net, and (3) intermediate layer of the combination module at the end of the U-net.

We designed two types of loss function for tumor segmentation and response prediction. To address the issue of class imbalance, we adopted the focal loss and used a sampling technique to ensure a constant ratio of the two classes in each training batch. Given the large number of parameters in our model, we employed several established strategies to minimize the risk of over-fitting, including cross-validation, data augmentation, instance normalization, and early stopping. We implemented the deep learning network on the open source TensorFlow platform with the Keras framework and Adam optimizer and trained using a NVIDIA Quadro P6000 GPU. Detailed description about the loss function and training procedures is provided in the Supplementary Methods.

**Network model ablation analysis**. To investigate whether different components of the proposed network are truly necessary for accurate response prediction, we simplified or modified the network and compared their performance (Supplementary Fig. 3). First, we removed the Siamese subnetwork and used a single ResNet to predict response based on pre-treatment images only. Second, we replaced the Siamese 3D-Unet architecture with Siamese ResNet and concatenated features from the final convolutional layers for the pre/post- treatment images. This network performs classification only, but no segmentation. Similar approaches were adopted for outcome prediction[33] and disease monitoring[10,32]. In addition, we designed an improved network where two Siamese subnetworks are joined at three different convolutional layers, similarly as our network. Again, this is a single-task learning network designed for classification only.

**Integration with blood markers**. We combined the imaging-based model with blood CEA levels to further improve accuracy of response prediction. First, CEA levels were dichotomized by using a well-established cutoff at 5 ng/ml for defining negative/positive samples. Based on dynamic information about pre/post-therapy blood CEA levels, we defined five mutually exclusive and exhaustive categories (Fig. 5a). These five categories coded as 1–5 were converted into dummy variables, i.e., binary indicator variables. The integrated model was developed in the training cohort using the random forest algorithm. The total number of trees (100) in the forest and maximum number of levels (3) in each decision tree were determined by using grid search with five-fold cross-validation in the training cohort.

**Evaluation of model performance**. We evaluated the accuracy of response prediction using receiver operating characteristic (ROC) analysis. The area under the ROC curve (AUC) was calculated to compare different models.

In addition, we computed sensitivity and specificity, and positive and negative predictive values. The optimal cutoff point was determined by maximizing the Youden's index on the ROC curve. The calibration curve was used to evaluate the prediction probability of the model.

**Statistical analysis**. Comparisons were performed using the $t$ test for continuous variables and Chi-square or Fisher's exact test for categorical variables, respectively. ROC curves were generated using the bootstrap method with 1000 replicates. The 95% confidence interval of AUCs was obtained using a non-parametric bootstrap approach by calculating the interval between 2.5 and 97.5 percentiles from the distribution of AUCs. The statistical significance for difference in AUC between models was assessed by the DeLong's test. The raw $p$ values for multiple pair-wise comparisons were adjusted using the Bonferroni correction by multiplying the number of tests. All statistical tests were two-sided and $p$ values below 0.05 were considered statistically significant.

**Reporting summary**. Further information on research design is available in the Nature Research Reporting Summary linked to this article.

## Data availability

The clinical and MRI data are not publicly available for patient privacy protection purposes. Any individual affiliated with an academic institution may request access to the original image and clinical data from the corresponding authors (X.W. and F.G.) for non-commercial, research purposes. Data will be provided with a signed data access agreement (Supplementary Fig. 14). Source data are provided with this paper.

## Code availability

The deep learning models were developed using standard libraries in open-source platforms including Keras and TensorFlow. Custom codes for the deployment of the model[38] are available https://github.com/Heng14/3D_RP-Net.

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

## Author contributions

C.J., F.G., and R.L. contributed to the conception and design of the study; C.J. and H.Y. developed the deep learning model; J.K., P.D., C.J. Y.Y., X.J., X.D., and J.T. contributed to acquisition and annotation of the data; C.J., J.K., D.T.C., and R.L. contributed to analysis and interpretation of the data; C.J. and R.L. wrote the initial manuscript; X.W., F.G., and R.L. supervised the work. All authors contributed to the revision of the manuscript.

## Competing interests

The Authors declare no competing interests.
