## [Peer Review File · Nature Communications]

Reviewers' Comments:

Reviewer #1:

Remarks to the Author:

The study proposed a deep learning strategy to provide tumor segmentation and predict complete response after neoadjuvant treatment in patients with rectal cancer.

Overall comments: The study were well done. The major strengths are internal validation using a prospective population, external validation in an independent population, and creation of an automatic segmentation which is critical for daily use. There are some

Introduction: The authors described the rationale and relevance of the study.

Methods:

- What was the inclusion and exclusion criteria of the study. I would like to see the flowchart with the patient accrual.
- I did not understand the subtopic ablation analysis.
- What is the meaning of "speed" image?
- Did the authors segmented all tumors? Including EMVI, nodes and tumor deposits?
- The authors included all sequences, including T1+C and DWI. However, the use of contrast media and DWI is not a consensus in the literature and it is not used in several centers. T2WI is the main sequence. Did the authors evaluated the performance of the features extracted only from the T2WI?

Results:

- I would suggest to include the p-values in the Table S1 to evaluate if the 3 population are balanced.
- In the table S1, some columns there is a space between the number and percentage and some columns not. Eg. Line 1 54(16.85%), 41(25.6%), and 44 (31.2%).
- It is not clear how the model to predict pathologic complete response using only the post treatment data performed compared with the combined one.
- What mesangial invasion mean in the figures9?
- The example of pCR showed in Fig 4 is a mucinous / colloid degeneration, which has high signal intensity on T2WI, differently from the other more common types of pCR, when we usually see fibrotic tissue (very low signal intensity on T2WI). How to balance that in the model?

Reviewer #2:

Remarks to the Author:

Interesting approach to designing a longitudinal data-based deep learning network for simultaneous segmentation/response prediction in rectal cancers.

- Is there a reason that the title doesn't actually mention rectal cancer?
- Using deep learning for longitudinal data has been explored recently, the novelty here appears to be in integrating the longitudinal information as well as the segmentation into a single system. The introduction should add a few lines to explain this better, currently the technical challenges in this regard is very sparse.
- The methods indicate that COMBAT was used for harmonization, in addition to other operations. Please comment on how this was done, as COMBAT transforms each "batch" of data and can require a variable of interest to be input. A quick skim of the code revealed a more standard "whitening" (mean subtraction, std dev division) as being used, was this done in addition to COMBAT?
- How was the tumor contoured in patients with pCR (i.e. no tumor)? Were the radiologists aware of pathologic outcome when annotating? Wouldn't the "tumor"ROI in such a case be naturally

smaller and less consistent -- which could be a confounding factor for the network? What was the kappa agreement between experts in annotation?

- Were the multiparametric images co-registered for inter-protocol differences?
 - Network description and training steps are well documented and explained
 - The categorization of blood CEA levels into 5 categories is unclear. Why do this? How was this information actually input to the random forest classifier? Why not use the CEA values themselves?
 - While confidence intervals are reported, how was significance assessed? Whether particular results were significant or not is not actually reported anywhere in the text, from what I could find.
 - Pg 7, Lines 7-8: "This indicates that adding the objective of predicting response did not compromise the effectiveness of tumor segmentation for the proposed network." -- weird line, since the objective was actually response prediction per the introduction?
 - Please note p-values and significant differences in all quantitative comparisons conducted.. the confidence intervals by themselves don't indicate significance.
 - In addition to excellent AUCs, analysis against alternative networks and as a function of location lend confidence in the generalizability of performance. Would have been curious to see if the performance is different between sexes, as pelvic anatomy changes.
 - The association of deep learning with the TRG is interesting, and again requires some kind of statistical analysis to determine the strength. This Fig S7 is one of the only places where it appears a significant difference is indicated between TRGs, but only for 1 comparison.
 - The attempt at interpreting the network output is commendable (assuming maps are accurately scaled with respect to each other). However, statements need more elaboration. How was it determined which layers corresponded to which protocols? Fig S8c is not intuitively associated with DWI to my eye.
 - Similarly in Fig S9, how were these specific attributes determined as being present? Were these responses consistently observed on radiology/pathology (?) in patients? Could this be quantified?
 - Similarly Fig 4 shows somewhat intuitive changes from pre- to post-, which again could be interesting to quantify if possible (especially given the Discussion point about "decrease")
 - While the final marker+model performance is excellent (with relevant specificity/sensitivity analysis), were there any trends in the few errors in the model?
 - The language in the Discussion could be toned down: "indispensable", "seamless" etc.
 - The statement on Pg 11, Lines 21-23: "One reason could be that tumor boundary information provided by segmentation enables the network to distinguish the primary tumor core from its invasive margin and surrounding peritumoral area" does not have any experiment associated with it. Please rephrase.
 - There have been a few papers on longitudinal data and deep learning recently, including DeepHit, MildInt etc, so statements on this aspect should be given appropriate context.
 - There have been several radiomics studies in rectal cancer with multi-institutional studies, these should be acknowledged. Several radiomics studies have also used pre- and post-treatment MRI, which also should be acknowledged.
- <https://ieeexplore.ieee.org/document/8681104>
<https://www.frontiersin.org/articles/10.3389/fgene.2019.00617/full>
<https://clincancerres.aacrjournals.org/content/23/23/7253.abstract> <-- pre and post
<https://www.nature.com/articles/s41467-020-18162-9>
- This is also an overstatement: "The radiomics approach relies on domain expertise to manually define hand-crafted features, which might lead to biased and incomplete characterization of the tumor" (Pg 12, Lines 24-25) since one could argue that deep learning approaches can overfit.
 - Pg 13, 8-9: "First, these networks were designed for the single task of risk prediction without explicit segmentation of the disease", why is explicit segmentation so important?
 - This statement should be rephrased, seems incomplete (Pg 13, 17-19): "Another distinction from the previous study is that only a single-center dataset was used, while we conducted rigorous internal and external validation of our deep learning model in a multi-institution study"
 - No limitations are noted, especially regarding the retrospective nature of the study, examining population differences etc.
 - Missing information: magnetic field strength.

Reviewer #3:

Remarks to the Author:

The authors focused on a hot topic of computational medical imaging analysis to predict pathological complete response after neoadjuvant chemoradiotherapy in rectal cancer patients. They presented a complex model to combine pre- and post-treatment images, but it seems that the model did not outperformed previously published models. Following are some concerns about this article:

1. There are now quite a few papers about pCR prediction with MRI based on radiomics or deep learning. It seems that the presented model did not achieved better prediction performance, so what are the main advantages of this method. The authors should compare the performance of the present method with that of previously published models.
2. From the example (Fig.4) given by the author, pCR patients have obvious differences between pre- and post-treatment images in the tumor area, which even could be easily visually obtained. Is it necessary to use such complicated methods for the prediction in clinical applications? There are several models based on less complex method with well performance, but do not rely on special computing devices.
3. "The tumor segmentation from the proposed network was in good agreement with the expert delineation with a mean Dice similarity coefficient of 0.87-0.90 for the two validation cohorts." Please listed a more clear and rigorous expression for the model performance of segmentation (e.g. mean [95% CI]). If the results of segmentation were not crucial, this part may move to the supplementary materials.
4. "When combined with blood-based tumor markers, the integrated model further improved prediction accuracy." It is necessary to express specific improvements in a numerical form directly in the abstract.
5. Experimental results need to be separated by more subheadings to help us locate the key results more clearly. The logic of the paper is very important.
6. "The deep learning score was consistently associated with four discrete categories of tumor regression grade." I was curious to see if some of the metrics commonly used by radiologists (e.g. size, radius, symmetry, and so on) were consistent with TRG in the data set.
7. Once a patient is predicted to become pCR, he or she may not receive any further treatment. In terms of safety, we need the model to have a PPV as high as possible, which is used to increase the confidence of clinical workers. But from the results given by the author, it seems that this risk is still inevitable.
8. The authors directly stacked images of four sequences as input data of the model. This could be inappropriate as different MRI sequences contained different information, and they should not be simply merged. The authors should take some further steps to process information of different sequences, which may also improve the performance of the model.
9. Which layers of the model do the visualization results come from? Deep learning models usually include a large number of feature layers. It may be interesting if the author provides visualization results of features from proposed depth-wise convolution layers, which was the biggest contributor for pCR prediction.
10. The model with post-treatment mp-MRI alone was also necessary for comparisons. This helps us clarify the contribution of the image before treatment and the reliability of the model for information fusion based on multi-time points mp-MRI.

AUTHORS' RESPONSE

We appreciate the thorough review and constructive comments and suggestions by the reviewers. We have revised the manuscript accordingly (highlighted changes), and the following is a point-by-point response to their comments.

REVIEWER COMMENTS

Reviewer #1 (Remarks to the Author):

The study proposed a deep learning strategy to provide tumor segmentation and predict complete response after neoadjuvant treatment in patients with rectal cancer.

Overall comments: The study were well done. The major strengths are internal validation using a prospective population, external validation in an independent population, and creation of an automatic segmentation which is critical for daily use.

Introduction: The authors described the rationale and relevance of the study.

Methods:

- What was the inclusion and exclusion criteria of the study. I would like to see the flowchart with the patient accrual.

Response: Thank you for this comment. We have clarified the study inclusion and exclusion criteria in the section on Methods: Study design and patients. The detailed flowchart for patient enrollment is shown in Figure S1.

- I did not understand the subtopic ablation analysis.

Response: For clarity, we have renamed this “Network model ablation analysis”.

This is a commonly used technique in deep learning, and the purpose is to evaluate how removing certain components or modules of a complex network would affect its performance. This will help determine whether different components of the proposed network are truly necessary for accurate response prediction, as compared to previously used simpler network models.

- What is the meaning of “speed” image?

Response: The speed image is a new synthetic image that is transformed from the original multi-parametric images. It has values in the range -1 to 1, i.e., the voxels inside the tumor having positive speed values, and non-tumor voxels having negative speed values. This has been clarified in the Supplementary Methods.

- Did the authors segmented all tumors? Including EMVI, nodes and tumor deposits?

Response: We segmented the gross tumor and invasive disease in surrounding tissue including EMVI. Regional lymph nodes were not contoured.

- The authors included all sequences, including T1+C and DWI. However, the use of contrast media and DWI is not a consensus in the literature and it is not used in several centers. T2WI is the main sequence. Did the authors evaluate the performance of the features extracted only from the T2WI?

Response: Thanks for the comment. We have trained a model based on the proposed network architecture using T2WI only. The results are shown in Table S4. The response prediction using T2WI was quite accurate (AUC 0.86-0.88) in the validation cohorts. But this did not outperform the model using multi-parametric imaging.

Results:

-I would suggest to include the p-values in the Table S1 to evaluate if the 3 population are balanced.

Response: We have added the P-values in the last column of Table 1.

-In the table S1, some columns there is a space between the number and percentage and some columns not. Eg. Line 1 54(16.85%), 41(25.6%), and 44 (31.2%).

Response: They have been corrected.

- It is not clear how the model to predict pathologic complete response using only the post treatment data performed compared with the combined one.

Response: We have trained a model using posttreatment image only. The results are shown in the updated Tables S2 and S3. As expected, this model did not perform nearly as well as the combined model.

- What mesangial invasion mean in the figureS9?

Response: This term was misused in our original manuscript. We have now corrected the term: “invasion of mesorectum” refers to tumor invasion into the perirectal fat that surrounds the rectum. This has been explained in the Figure.

- The example of pCR showed in Fig 4 is a mucinous / colloid degeneration, which has high signal intensity on T2WI, differently from the other more common types of pCR, when we usually see fibrotic tissue (very low signal intensity on T2WI). How to balance that in the model?

Response: Thank you for this insightful comment. Since the two types of response patterns you mentioned are both included in our training set, the deep neural network has the capacity to learn the appropriate representations for the different patterns on T2WI and can still produce the correct prediction. Additionally, while the model takes into account the signal intensity, other

information such as changes in tumor morphology, heterogeneity, and invasion to surrounding tissue is also used in the prediction. Moreover, the model does not rely on T2WI only, it also considers complementary information from other images such as T1W and DWI. Our results on comparing T2WI and multiparametric imaging (Table S8) confirmed this.

Reviewer #2 (Remarks to the Author):

Interesting approach to designing a longitudinal data-based deep learning network for simultaneous segmentation/response prediction in rectal cancers.

- Is there a reason that the title doesn't actually mention rectal cancer?

Response: We intend to present this work as a general method to extract dynamic information and predict response from longitudinal images, which can be applied to many cancer types and other disease. We think this is an underexplored area in deep learning. It is our hope that this study can bring the attention of more researchers to tackle this important problem.

- Using deep learning for longitudinal data has been explored recently, the novelty here appears to be in integrating the longitudinal information as well as the segmentation into a single system. The introduction should add a few lines to explain this better, currently the technical challenges in this regard is very sparse.

Response: Thank you for this comment. We have revised the introduction.

- The methods indicate that COMBAT was used for harmonization, in addition to other operations. Please comment on how this was done, as COMBAT transforms each "batch" of data and can require a variable of interest to be input. A quick skim of the code revealed a more standard "whitening" (mean subtraction, std dev division) as being used, was this done in addition to COMBAT?

Response: The main purpose of COMBAT was to remove differences across the two institutions. This was done separately for each MRI sequence, with the institution (internal or external) as the variable of interest, i.e., batch. Yes, normalization was done in addition to COMBAT.

- How was the tumor contoured in patients with pCR (i.e. no tumor)? Were the radiologists aware of pathologic outcome when annotating? Wouldn't the "tumor"ROI in such a case be naturally smaller and less consistent -- which could be a confounding factor for the network? What was the kappa agreement between experts in annotation?

Response: Radiologists contoured the abnormal anatomic structures including any suspicious residual tumor and treatment-induced changes in post-therapy images. And they were blinded to the pathologic outcome when contouring. This has been clarified in the Supplementary Methods. You are correct that, on average, the contoured tumors in the pCR group were smaller than non-pCR post treatment. However, there was significant variability and overlap in the change of tumor volume between TRG0 (pCR) and TRG1 (non-pCR) (Fig. S8). A model based on the change in tumor volume alone was not sufficiently accurate for predicting pCR (AUC: 0.72-0.79 in validation cohorts). This suggests that the deep learning model uses not only tumor contour, but also additional information contained in the images.

- Were the multiparametric images co-registered for inter-protocol differences?

Response: Yes, they were co-registered before input to the model. Please refer to Supplementary Methods: Image processing.

- Network description and training steps are well documented and explained

- The categorization of blood CEA levels into 5 categories is unclear. Why do this? How was this information actually input to the random forest classifier? Why not use the CEA values themselves?

Response: We first divided CEA levels by using a well-established cutoff at 5 ng/ml for defining negative/positive samples. Then we defined five mutually exclusive and exhaustive categories based on varying degree of response according to the dynamic pattern of CEA levels.

The 5 discrete categories were converted into dummy variables (i.e., binary indicator variables) and input the random forest model.

We used the continuous CEA level in the combined model and the performance was slightly worse than the model with 5 categories (Fig. S13).

- While confidence intervals are reported, how was significance assessed? Whether particular results were significant or not is not actually reported anywhere in the text, from what I could find.

Response: The statistical significance for difference in AUC between models was assessed by the DeLong's test. This has been clarified in Methods: Evaluation of model performance.

- Pg 7, Lines 7-8: "This indicates that adding the objective of predicting response did not compromise the effectiveness of tumor segmentation for the proposed network." -- weird line, since the objective was actually response prediction per the introduction?

Response: We have removed this sentence.

- Please note p-values and significant differences in all quantitative comparisons conducted. the confidence intervals by themselves don't indicate significance.

Response: Thanks for the comment. We have reported the p values for all quantitative comparisons.

- In addition to excellent AUCs, analysis against alternative networks and as a function of location lend confidence in the generalizability of performance. Would have been curious to see if the performance is different between sexes, as pelvic anatomy changes.

Response: We have reported the prediction performance as stratified by sex. The results are shown in Tables S9 and S10. The performance was slightly better in male sex.

- The association of deep learning with the TRG is interesting, and again requires some kind of statistical analysis to determine the strength. This Fig S7 is one of the only places where it appears a significant difference is indicated between TRGs, but only for 1 comparison.

Response: We have reported the p values for the following comparisons between TRG0 vs TRG1; TRG1 vs TRG2; TRG2 vs TRG3.

- The attempt at interpreting the network output is commendable (assuming maps are accurately scaled with respect to each other). However, statements need more elaboration. How was it determined which layers corresponded to which protocols? Fig S8c is not intuitively associated with DWI to my eye.

Response: This is based on visual observation of the feature maps at different layers and different modalities of MRI as well as a general understanding of deep convolutional neural networks as applied to image analysis. We do not claim there is a one-to-one correspondence between the two. We have revised this part as below:

“The image features at shallow layers mainly reflect the structural information, such as tumor boundary, shape, and texture, based on T1w and T2w MRI. On the other hand, features at deep layers mainly represent high-level semantic tumor characteristics from anatomical images as well as functional information contained in DWI.”

- Similarly in Fig S9, how were these specific attributes determined as being present? Were these responses consistently observed on radiology/pathology (?) in patients? Could this be quantified?

Response: These attributes were determined mainly based on the radiologist interpretation of the multi-parametric MRI scans in relation to the corresponding feature maps. The patterns were generally consistent across patients. This figure shows 5 representative cases (each column is a patient).

Yes, the feature map is quantitative. We have added a side bar to show the color scale.

- Similarly Fig 4 shows somewhat intuitive changes from pre- to post-, which again could be interesting to quantify if possible (especially given the Discussion point about "decrease")

Response: Thank you for the comment. A side bar has been added to show the color scale. We computed the difference in magnitude between feature maps in the 201st channel, which showed 88.9% decrease from pre- to post-CRT image for the pCR case; but only 11.3% decrease for the non-pCR case.

- While the final marker+model performance is excellent (with relevant specificity/sensitivity analysis), were there any trends in the few errors in the model?

Response: We compared the sensitivity, PPV, and NPV between different models at specificity thresholds of 95% and 99%. As shown in Figure 5, the integrated model significantly improved sensitivity and NPV value at both thresholds and PPV at 99% specificity ($p < 0.01$).

- The language in the Discussion could be toned down: "indispensable", "seamless" etc.

Response: These words have been removed.

- The statement on Pg 11, Lines 21-23: "One reason could be that tumor boundary information provided by segmentation enables the network to distinguish the primary tumor core from its invasive margin and surrounding peritumoral area" does not have any experiment associated with it. Please rephrase.

Response: This has been revised.

- There have been a few papers on longitudinal data and deep learning recently, including DeepHit, MildInt etc, so statements on this aspect should be given appropriate context.

Response: Thanks for pointing this out. we have revised it and included these references.

<https://ieeexplore.ieee.org/document/8681104>

<https://www.frontiersin.org/articles/10.3389/fgene.2019.00617/full>

- There have been several radiomics studies in rectal cancer with multi-institutional studies, these should be acknowledged. Several radiomics studies have also used pre- and post-treatment MRI, which also should be acknowledged.

Response: In our original manuscript, the 2017 study was already included, which used a single-institutional dataset. The 2020 study did use multi-institutional datasets but for a different clinical application, *i.e.*, predicting distant metastasis after surgery by radiomic analysis of pre-treatment MRI (T2WI and DWI). We have included this in our Discussion.

<https://clincancerres.aacrjournals.org/content/23/23/7253.abstract> <-- pre and post

<https://www.nature.com/articles/s41467-020-18162-9>

- This is also an overstatement: "The radiomics approach relies on domain expertise to manually define hand-crafted features, which might lead to biased and incomplete characterization of the tumor" (Pg 12, Lines 24-25) since one could argue that deep learning approaches can overfit.

Response: This sentence has been revised.

- Pg 13, 8-9: "First, these networks were designed for the single task of risk prediction without explicit segmentation of the disease", why is explicit segmentation so important?

Response: We compared with network models that do not perform explicit tumor segmentation (Fig. S4). The results are worse than proposed model as shown in Table S2 and S3.

- This statement should be rephrased, seems incomplete (Pg 13, 17-19): "Another distinction from

the previous study is that only a single-center dataset was used, while we conducted rigorous internal and external validation of our deep learning model in a multi-institution study"

Response: This has been revised.

- No limitations are noted, especially regarding the retrospective nature of the study, examining population differences etc.

Response: Thank you for pointing this out. We have added a paragraph to discuss the study limitations.

- Missing information: magnetic field strength.

Response: The information about magnetic field strength for each cohort has been added in Table S1.

Reviewer #3 (Remarks to the Author):

The authors focused on a hot topic of computational medical imaging analysis to predict pathological complete response after neoadjuvant chemoradiotherapy in rectal cancer patients. They presented a complex model to combine pre- and post-treatment images, but it seems that the model did not outperformed previously published models. Following are some concerns about this article:

1. There are now quite a few papers about pCR prediction with MRI based on radiomics or deep learning. It seems that the presented model did not achieved better prediction performance, so what are the main advantages of this method. The authors should compare the performance of the present method with that of previously published models.

Response: The methodological advance of our approach over previous studies has been discussed in detail in the Discussion (Line 273-297). Here, we mention them briefly and highlight a few key points.

A number of studies have used radiomics or deep learning to predict pathologic response to neoadjuvant CRT in rectal cancer. However, most of these studies included small, single-institution datasets, using only pre-treatment images. The advantage of incorporating dynamic information in pre- and post-treatment images for response prediction has been clearly demonstrated in our study (Fig. 3, Table S2, S3).

To maximally extract information from longitudinal images, we proposed a novel multi-task deep neural network that performs simultaneously tumor segmentation and response prediction. In addition, we designed the network architecture by combining feature representations from multiple layers, which allowed for multi-scale integration and in-depth comparison of paired images. Through extensive experiments for model evaluation and benchmark comparison, our approach demonstrated superior prediction performance compared with alternative deep learning models. The following series of experiments were conducted in this study, with detailed results presented in the main article or supplementary material.

- i. Model based on the change in tumor volume between Pre and Post treatment;
- ii. Resnet-18 model based on pre-treatment multi-parametric MRI;
- iii. Resnet-18 model based on post-treatment multi-parametric MRI;
- iv. Traditional Siamese Resnet-18 model based on pre and post-treatment multi-parametric MRI;
- v. Multi-scale Siamese Resnet-18 model based on pre and post-treatment multi-parametric MRI;
- vi. 3D RP-Net (Pre+Post) based on pre and post-treatment T2W MRI;
- vii. 3D RP-Net (Pre+Post) with integration of blood-based biomarkers;
- viii. 3D RP-Net (Pre+Post) for predicting response at different tumor locations;
- ix. 3D RP-Net (Pre+Post) for predicting response in different genders.

Two previous studies (Liu, 2017; Zhang 2020) used pre-treatment and post-treatment images for response prediction in rectal cancer. Both studies included single-institution data and reported AUC >0.97 in an internal validation cohort of 70 and 93 patients, respectively. On the other hand, we used multi-institution data for training and validation, and our deep learning model achieved similarly excellent performance (AUC=0.95 and 0.92) in the internal and external validation cohorts of 160 and 141 patients.

We caution that the performance between studies is not directly comparable because of different study design (single-institution vs multi-institution), the use of different patient cohorts/datasets, methodologies, and MRI sequences. Of note, the 2020 study adopted a novel MRI technology, diffusion kurtosis imaging, which is not widely used in clinical settings. Therefore, their models cannot be directly applied to our data, which used standard MRI sequences.

Reference

1. Liu, Zhenyu, et al. "Radiomics analysis for evaluation of pathological complete response to neoadjuvant chemoradiotherapy in locally advanced rectal cancer." *Clinical Cancer Research*. 23.23 (2017): 7253-7262.
2. Zhang, Xiao-Yan, et al. "Predicting rectal cancer response to neoadjuvant chemoradiotherapy using deep learning of diffusion kurtosis MRI." *Radiology*. (2020): 190936.

2. From the example (Fig.4) given by the author, pCR patients have obvious differences between pre- and post-treatment images in the tumor area, which even could be easily visually obtained. Is it necessary to use such complicated methods for the prediction in clinical applications? There are several models based on less complex method with well performance, but do not rely on special computing devices.

Response: These examples are shown only for illustration purposes to highlight the changes. In reality, however, there are many other cases with complex tumor response patterns and treatment-induced changes in the surrounding normal tissue that could make visual assessment unreliable. For highly accurate prediction, this will require a sophisticated model as demonstrated in our comparison with other simpler models.

3. "The tumor segmentation from the proposed network was in good agreement with the expert delineation with a mean Dice similarity coefficient of 0.87-0.90 for the two validation cohorts." Please listed a more clear and rigorous expression for the model performance of segmentation (e.g. mean [95% CI]). If the results of segmentation were not crucial, this part may move to the supplementary materials.

Response: We have moved these results into the supplementary Figure S3.

4. “When combined with blood-based tumor markers, the integrated model further improved prediction accuracy.” It is necessary to express specific improvements in a numerical form directly in the abstract.

Response: This is revised as suggested.

5. Experimental results need to be separated by more subheadings to help us locate the key results more clearly. The logic of the paper is very important.

Response: Thanks for the suggestion. We have included more section subheadings.

6. “The deep learning score was consistently associated with four discrete categories of tumor regression grade.” I was curious to see if some of the metrics commonly used by radiologists (e.g. size, radius, symmetry, and so on) were consistent with TRG in the data set.

Response: We analyzed the relationship between TRG and the relative change in tumor volume in all the cohorts. While there was a trend across TRG, the distinction between TRG0 (pCR) and TRG1 (non-pCR) was not sufficiently clear based on tumor volume (Fig. S8). By contrast, these two groups were well separated based on the deep learning model (Fig. S9).

7. Once a patient is predicted to become pCR, he or she may not receive any further treatment. In terms of safety, we need the model to have a PPV as high as possible, which is used to increase the confidence of clinical workers. But from the results given by the author, it seems that this risk is still inevitable.

Response: This is an important point and we fully agree with you. We have discussed this as a limitation.

8. The authors directly stacked images of four sequences as input data of the model. This could be inappropriate as different MRI sequences contained different information, and they should not be simply merged. The authors should take some further steps to process information of different sequences, which may also improve the performance of the model.

Response: We agree with you that different MRI sequences contain different information and that they should be treated differently. In this work, we did not simply stack the images together, but instead used a 4D tensor image as input. Specifically, each image is resized to $256 \times 256 \times 16$ so the input data size is $256 \times 256 \times 16 \times 4$, where 4 means four sequences. During the training process, the network learns different 3D weights for each channel so that different information is captured for each MRI sequence. We originally indicated this in the Supplementary Methods but now have updated the Methods to clarify this point.

“Rather than simply stacking the multi-parametric images together, we combined them in a 4-dimensional tensor image with the last dimension being each of the 4 MRI sequences and fed this as input to the network model.”

9. Which layers of the model do the visualization results come from? Deep learning models usually include a large number of feature layers. It may be interesting if the author provides visualization results of features from proposed depth-wise convolution layers, which was the biggest contributor for pCR prediction.

Response: The visualization of feature maps was indeed from the depth-wise convolution layers. This has been clarified.

10. The model with post-treatment mp-MRI alone was also necessary for comparisons. This helps us clarify the contribution of the image before treatment and the reliability of the model for information fusion based on multi-time points mp-MRI.

Response: We have trained a model using post-treatment mp-MRI alone. The results are shown in the updated Tables S2 and S3. As expected, this model did not perform nearly as well as the combined model.

Reviewers' Comments:

Reviewer #1:

Remarks to the Author:

The authors addressed the issues that were raised accordingly.

Reviewer #2:

Remarks to the Author:

In general, this is a much improved version of the paper with a majority of the comments addressed. There are some areas that need clarification before I can recommend acceptance.

- The transformation of the original multi-parametric image to a speed image is still unclear. What does the transformation entail? Why does this image have to be generated and included alongside the original images?
- What do the p-value now included in Table 1 represent? Are the authors saying that the 3 cohorts were significantly different in terms of all the clinical parameters? That would imply that the cohorts are fundamentally non-comparable?
- Was agreement between experts computed for the annotations?
- Fig S13 helps clarify my question regarding CEA levels. The authors need to be consistent in reporting p-value significance in ALL such comparisons as this helps the reader understand which trends are most important.
- Fig S11 would benefit from the original MRIs being shown as well.
- S10, S11 are better explained but incompletely. Fig S10 appears to be a general statement about how activation maps work, though how many datasets were included in this observation are unclear. Fig S11 similarly mentions specific layers that were "consistently observed across patients", maybe mention how many patients these specific layers were activated in? Again, are all of these images/observations being drawn from training or testing? There is no mention in the text that these observations are based on radiologist evaluation of the scans and maps. All of this needs to be better detailed: only XX channels show high activation, these were evaluated by radiologist in conjunction with MP-MRI, these layers found to capture these physiologic characteristics etc.
- Fig 4 is using the same layers from S11 (referred to now as "key channels", but only 1 trend is reported (submucosal lesions). Similar trends should be reported in the figure/caption for all the layers.

Reviewer #3:

Remarks to the Author:

Thanks for the work of the authors to revise the manuscript. A part of my issues were well addressed. But there are still some major concerns.

1. Compared with previous studies, the present manuscript doesnot show advantages in AUC and even disadvantages in PPV. Although the authors mentioned these as limitations, it cannot be ignored. Such a complex model was constructed based on multi-sequences and multi-time points data, but it did not bring any improvement in performance. It is possible to compare the performance of the proposed method with that of previous studies. There were some software (e.g. PyRadiomics) to construct radiomics model which were used in previous studies.
2. All of the present manuscript and previous studies showed that imaging analysis can predict pCR accurately based on retrospective clinical data. It is prospective verification that is in need to further confirm the confidence of the radiomics or deep learning model in clinical application.
3. How to overcome the influence of different parameters of MRI acquisition in the research, such as field strength (3T and 1.5T)? What about the performance if the cohort were divided according

to the field strength?

4. The author built the input of the model through data stacking, more details of registration of 2-D image space and slice alignment between different sequences of MRI should be provided, especially in T1W+C and DWI. Did the authors try more elegant information fusion strategies?

Response to Reviews' Comments

Reviewer #2 (Remarks to the Author):

In general, this is a much-improved version of the paper with a majority of the comments addressed. There are some areas that need clarification before I can recommend acceptance.

-The transformation of the original multi-parametric image to a speed image is still unclear. What does the transformation entail? Why does this image have to be generated and included alongside the original images?

Reply: First, we clarify that the sole purpose of the speed image is to help generate a **manual** delineation of the tumor. It is not used or needed for the prediction of deep learning model. Second, the transformation to a speed image is a preprocessing procedure in which the tumor in the original multi-parametric image is segmented using a clustering algorithm in the ITK-SNAP software. Based on this initial segmentation, a radiologist then reviewed and fine-tuned the tumor region to generate a more precise tumor delineation. This has been clarified in the Supplementary Methods.

- What do the p-value now included in Table 1 represent? Are the authors saying that the 3 cohorts were significantly different in terms of all the clinical parameters? That would imply that the cohorts are fundamentally noncomparable?

Reply: The p-values are for comparing the 3 cohorts. Because patients in our study were not recruited from a randomized controlled trial, some differences in the distribution are expected. This kind of pattern is very common in retrospective studies based on observational data. We emphasize that all 3 cohorts had the same enrollment criteria as described in the Methods. All eligible patients were included in the study. Importantly, all patients were diagnosed with locally advanced rectal cancer and treated with neoadjuvant chemoradiotherapy followed by total mesorectal excision. As such, they represent similar patient populations (but not identical).

- Was agreement between experts computed for the annotations?

Reply: Yes. We computed the agreement between expert contours for all tumors, and the Dice coefficient was 0.92 ± 0.03 (mean, standard deviation). This has been added to the Figure S3.

- Fig S13 helps clarify my question regarding CEA levels. The authors need to be consistent in reporting p-value significance in ALL such comparisons as this helps the reader understand which trends are most important.

Reply: We have updated this information in the figure caption.

- Fig S11 would benefit from the original MRIs being shown as well.

Reply: We added a row to show the original MRI in Figure S11.

- S10, S11 are better explained but incompletely. Fig S10 appears to be a general statement about how activation maps work, though how many datasets were included in this observation are unclear. Fig S11 similarly mentions specific layers that were "consistently observed across patients", maybe mention how many patients these specific layers were activated in? Again, are all of these images/observations being drawn from training or testing? There is no mention in the text that these observations are based on radiologist evaluation of the scans and maps. All of this needs to be better detailed: only XX channels show high activation, these were evaluated by radiologist in conjunction with MP-MRI, these layers found to capture these physiologic characteristics etc.

Reply: Thanks for the comment. We have clarified these points in the text and the figure captions.

- Fig. 4 is using the same layers from S11 (referred to now as "key channels", but only 1 trend is reported (submucosal lesions). Similar trends should be reported in the figure/caption for all the layers.

Reply: We have added the detailed information in the figure caption for Fig. 4.

Reviewer #3 (Remarks to the Author):

1. Compared with previous studies, the present manuscript does not show advantages in AUC and even disadvantages in PPV. Although the authors mentioned these as limitations, it cannot be ignored. Such a complex model was constructed based on multi-sequences and multi-time points data, but it did not bring any improvement in performance. It is possible to compare the performance of the proposed method with that of previous studies. There were some software (e.g. PyRadiomics) to construct radiomics model which were used in previous studies.

Reply: Thank you for your suggestion. We have constructed a radiomics model for pCR prediction using the PyRadiomics software. The detailed method and results are included in the revision and also shown here.

Radiomics model for pCR prediction

Radiomics features were extracted from both pre-CRT and post-CRT multiparametric MRI using the PyRadiomics software package. For each MRI sequence, a total of 1074 3D radiomics features were extracted: 1) 271 first-order features reflecting the distribution of voxel intensities within tumor region; 3) 803 texture features: including computational features based on gray level co-occurrence matrix (n=351): gray level run length matrix (n=226), and gray level size zone matrix (n=226). These features quantitatively describe the higher-order voxel-level heterogeneity in the tumor. In addition, 21 3D shape features describing the tumor morphological and structural features were calculated. In training set (321 cases of MRIs after nCRT), the optimal subset of radiomics features associated with the response of nCRT (pCR and non-pCR) was selected by applying the minimum-redundancy-maximum-relevance technique. Finally, we built a logistic regression model combined with the least absolute shrinkage and selection operator (LASSO) to predict the response and tested the performance in the validation cohorts.

Table S5. Detailed information for prediction performance of the radiomics model in the study cohorts.

Cohort	Accuracy (95% CI)	AUROC (95% CI)	Sensitivity (95% CI)	Specificity (95% CI)	PPV (95% CI)	NPV (95% CI)
Training	90.97% (87.29%-93.67%)	0.931 (0.901-0.962)	89.47% (77.81%-95.65%)	91.29% (87.05%-94.28%)	68.92% (56.96%-78.89%)	97.57% (94.53%-99.01%)
Internal validation	83.75% (77.20%-88.72%)	0.889 (0.838-0.940)	84.09% (69.33%-92.84%)	83.62% (75.35%-89.61%)	66.07% (52.09%-77.84%)	93.27% (86.15%-97.02%)
External validation	80.85% (73.52%-86.54%)	0.860 (0.796-0.924)	79.07% (63.52%-89.42%)	81.63% (72.26%-88.47%)	65.38% (50.84%-77.67%)	89.89% (81.21%-94.98%)

Note: PPV, positive predictive value; NPV, negative predictive value.

2. All of the present manuscript and previous studies showed that imaging analysis can predict pCR accurately based on retrospective clinical data. It is prospective verification that is in need to further confirm the confidence of the radiomics or deep learning model in clinical application.

Reply: We totally agree with you on this point. We have discussed this in detail as limitations of the study.

Please refer to the third last paragraph in Discussion.

‘First, it is a retrospective study and subject to potential selection bias. The generalizability and clinical utility of the proposed model should be rigorously tested in future prospective studies.’

‘This deep learning model is not yet ready for clinical use, given the need for prospective validation and demonstration of a sufficiently high positive predictive value.’

3. How to overcome the influence of different parameters of MRI acquisition in the research, such as field strength (3T and 1.5T)? What about the performance if the cohort were divided according to the field strength?

Reply: Our strategy to overcome the influence of imaging parameters is based on data augmentation, which is widely used for training deep neural networks. This technique increases the number and diversity of training samples and has been shown to improve model generalizability. Specifically, the data augmentation procedures include random image rotation and shifts, image smoothing and enhancement with Gaussian filtering and Laplacian filtering, as well as adding image noise.

We also performed subgroup analysis based on the magnetic field strength, and the results for internal and external validation cohorts are shown in Table S10-11, respectively. There was a slight improvement in prediction performance (around 2-4% increase in AUC) in patients scanned under a magnetic field strength of 3T compared with 1.5T. This is possibly because of the improved image quality in 3T MRI.

4. The author built the input of the model through data stacking, more details of registration of 2-D image space and slice alignment between different sequences of MRI should be provided, especially in T1W+C and DWI. Did the authors try more elegant information fusion strategies?

Reply: We first performed 3D rigid image registration (translation and rotation) between T1-weighted and T2-weighted MRI using normalized mutual information as the loss function. We then registered the diffusion-weighted image to the T2-weighted image with affine transformation in order to correct for the eddy current distortion and motion effects. Finally, all MRI sequences were aligned to the same spatial coordinates with T2-weighted image as the reference. Registration was implemented using the extensively benchmarked elastix software in ITK. The registered images were visually checked for quality assurance by the radiologist. This has been clarified in the Supplementary Methods.

In this work, we fed the multiparametric MRI data as a 4D tensor image. This approach allows voxel-level integration of image data and is a widely used and established strategy for multimodality image fusion. Additional information fusion occurs in the different layers of the convolutional neural network. We believe the image data are being effectively mined with this approach, which produced highly accurate prediction. Further improvement in prediction performance will most likely come from incorporating different types of information other than imaging. As mentioned in the Discussion, it will be important to integrate information from other investigations such as clinical exam, endoscopic assessment, or molecular approaches to further improve the prediction accuracy.

Reviewers' Comments:

Reviewer #2:

Remarks to the Author:

The updated manuscript is much improved, and a lot of the details I had requested have now been provided -- especially the speed image, Dice coefficient, p-values etc. The additional comparison against radiomics further bolsters a very strong paper. I think this is nearly ready for acceptance, pursuant to making the edits below (which are not very laborious).

The biggest concern I have remains with Table 1, which presents the information poorly. The goal with such a table is to demonstrate if any of the variables in that table are significantly different or biased between response groups. Instead it summarizes the proportion of patients who are male, female, upper tumors, lower tumors etc in each of the 3 cohorts -- which would obviously be significantly different. This is not informative. Each cohort should be split by response group (i.e. pCR, non-pCR) and the entries would then be how many males are pCR/non-pCR in each of discovery, int validation, ext validation; and so on. The significance testing would be to determine if any specific variable in terms of sex, tumor location, etc is associated with pCR i.e. in the discovery cohort, are males significantly more likely to achieve pCR?

Excellent examples are in:

Table 1, <https://clincancerres.aacrjournals.org/content/23/23/7253>

Table 2, <https://doi.org/10.1002/jmri.27140>

Figure S13: Please take out the statement "However, there was a trend between the imaging model and integrated model with 3D RP-Net + CEA' ($p = 0.07$)". Statistical testing shows something is significant or it isn't based on meeting a p-value cutoff, it cannot be used to "suggest" significance based on being "close to a p-value cutoff".

A minor point here (which I may have forgotten to note in previous reviews) is that since multiple pairwise comparisons have been performed in each experiment, the p-value threshold should be adjusted via Bonferroni correction. So the cutoff when doing 4 pairwise comparisons such as in Figure S13 would be $0.05/4 = 0.012$. Please adjust the significance threshold for ALL pairwise comparisons appropriately.

Pg 8, Line 138-146: Please appropriately note that a comparison against radiomics was performed, and where in the supplementary materials this information is provided. Otherwise, Line 151 is the first time a reference is made to comparing against radiomics.

Reviewer #3:

Remarks to the Author:

The authors addressed the issues.

The updated manuscript is much improved, and a lot of the details I had requested have now been provided -- especially the speed image, Dice coefficient, p-values etc. The additional comparison against radiomics further bolsters a very strong paper. I think this is nearly ready for acceptance, pursuant to making the edits below (which are not very laborious).

The biggest concern I have remains with Table 1, which presents the information poorly. The goal with such a table is to demonstrate if any of the variables in that table are significantly different or biased between response groups. Instead it summarizes the proportion of patients who are male, female, upper tumors, lower tumors etc in each of the 3 cohorts -- which would obviously be significantly different. This is not informative. Each cohort should be split by response group (i.e. pCR, non-pCR) and the entries would then be how many males are pCR/non-pCR in each of discovery, int validation, ext validation; and so on. The significance testing would be to determine if any specific variable in terms of sex, tumor location, etc is associated with pCR i.e. in the discovery cohort, are males significantly more likely to achieve pCR?

We have revised Table 1 as suggested.

Figure S13: Please take out the statement "However, there was a trend between the imaging model and integrated model with 3D RP-Net + CEA' ($p = 0.07$)". Statistical testing shows something is significant or it isn't based on meeting a p-value cutoff, it cannot be used to "suggest" significance based on being "close to a p-value cutoff".

We have removed this statement.

A minor point here (which I may have forgotten to note in previous reviews) is that since multiple pairwise comparisons have been performed in each experiment, the p-value threshold should be adjusted via Bonferroni correction. So the cutoff when doing 4 pairwise comparisons such as in Figure S13 would be $0.05/4 = 0.012$. Please adjust the significance threshold for ALL pairwise comparisons appropriately.

We have corrected all the p values for multiple comparisons by Bonferroni correction. As the number of tests vary in different scenarios, we adjusted the raw p values by multiplying the number of comparisons. We also clarified this in Methods.

Pg 8, Line 138-146: Please appropriately note that a comparison against radiomics was performed, and where in the supplementary materials this information is provided. Otherwise, Line 151 is the first time a reference is made to comparing against radiomics.

We have updated the sentence.

Reviewers' Comments:

Reviewer #2:

Remarks to the Author:

Great job addressing all my comments. No further concerns or issues.